# Myeloma Bone Disease: The Osteoblast in the Spotlight

**DOI:** 10.3390/jcm10173973

**Published:** 2021-09-02

**Authors:** Rebecca E. Andrews, Janet E. Brown, Michelle A. Lawson, Andrew D. Chantry

**Affiliations:** 1Department of Oncology and Metabolism, The Medical School, The University of Sheffield, Sheffield S10 2RX, UK; j.e.brown@sheffield.ac.uk (J.E.B.); m.a.lawson@sheffield.ac.uk (M.A.L.); a.d.chantry@sheffield.ac.uk (A.D.C.); 2Department of Haematology, Sheffield Teaching Hospitals NHS Foundation Trust, Royal Hallamshire Hospital, Sheffield S10 2JF, UK

**Keywords:** myeloma bone disease, osteoblast, bone anabolic, osteolytic lesions, multiple myeloma, osteogenesis

## Abstract

Lytic bone disease remains a life-altering complication of multiple myeloma, with up to 90% of sufferers experiencing skeletal events at some point in their cancer journey. This tumour-induced bone disease is driven by an upregulation of bone resorption (via increased osteoclast (OC) activity) and a downregulation of bone formation (via reduced osteoblast (OB) activity), leading to phenotypic osteolysis. Treatments are limited, and currently exclusively target OCs. Despite existing bone targeting therapies, patients successfully achieving remission from their cancer can still be left with chronic pain, poor mobility, and reduced quality of life as a result of bone disease. As such, the field is desperately in need of new and improved bone-modulating therapeutic agents. One such option is the use of bone anabolics, drugs that are gaining traction in the osteoporosis field following successful clinical trials. The prospect of using these therapies in relation to myeloma is an attractive option, as they aim to stimulate OBs, as opposed to existing therapeutics that do little to orchestrate new bone formation. The preclinical application of bone anabolics in myeloma mouse models has demonstrated positive outcomes for bone repair and fracture resistance. Here, we review the role of the OB in the pathophysiology of myeloma-induced bone disease and explore whether novel OB targeted therapies could improve outcomes for patients.

## 1. Introduction

Multiple myeloma (MM) is the second most common haematological malignancy and accounts for approximately 1% of all cancer diagnoses worldwide [1,2]. This B-cell malignancy is characterised by the clonal expansion of malignant plasma cells (MPCs) (>10%) within the bone marrow (BM) (or confirmation plasmacytoma) and is normally associated with a measurable monoclonal immunoglobulin secreted from the MPCs [3]. MM causes hypercalcaemia, renal failure, anaemia, and bone disease (hence its diagnostic acronym, “CRAB criteria”). Predictive modelling suggests that MM incidence will continue to rise, in part due to the effects of an aging population [4], with a predicted increase of 11% from 2014 to 2035 [5]. Of all patients diagnosed with MM, 80–90% will develop the associated bone disease during their cancer journey [6,7]. This leads to losses in trabecular bone, reduced bone mineral density (BMD), and phenotypic osteolytic lesions. The risk of fracture for these patients increases nine-fold [8], and those who do suffer a pathological fracture have an increased mortality of 20% compared to those without a fracture [9]. The clinical impact of myeloma bone disease (MBD) is significant, leading to bone pain, poor mobility, and, subsequently, decreased quality of life.

## 2. Pathophysiology of Myeloma Bone Disease

Bone health is maintained throughout one’s life by the process of bone remodelling, a continuous process in which osteoclasts (OCs) remove old bone, and osteoblasts (OBs) lay down new bone. Bone remodelling is orchestrated by osteocytes (OCYs) in biomechanical response to loading forces or microdamage to the bone. Many signalling pathways are in place to maintain the bone remodelling equilibrium. The most important identified pathway is that of the receptor activator of nuclear factor kappa B (RANK)/RANK ligand (RANKL)/Osteoprotegerin (OPG) signalling (Figure 1). When RANKL (produced by both OBs and OCYs) binds to RANK on the surface of immature OCs, it leads to OC differentiation to mitigate mature OCs to resorb bone. This is moderated by the release of OPG, also produced by OBs, which acts as a decoy receptor to RANKL to reduce OC formation/activity. In the case of MM, the presence of MPCs and their interaction with bone marrow stromal cells (BMSCs) causes bone remodelling to become dysregulated. In brief, MBD is driven by (a) an upregulation of OC activity, (b) an inhibition of OB activity, and (c) positive feedback, which accelerates tumour growth (Figure 1). This results in net bone loss, profound osteolysis, losses in trabecular structure, and compromised skeletal strength and function [10]. The change from healthy bone to MBD is driven by MPCs in the bone marrow microenvironment (BMME) and subsequent alterations in cytokines, extracellular vesicles, and cellular signalling/interactions. Factors driving osteolysis are often categorised as either OC-activating factors (OAFs) or OB-inhibiting factors (OIFs; Table 1).

OAFs (e.g., RANKL, macrophage inflammatory protein-1 α (MIP-1α), and Activin A) are directly expressed by MPCs or indirectly increased by MPC signalling to other cells in the BMME. The upregulation of bone resorption results in a release of growth factors from the resorbed bone matrix (e.g., transforming growth factorβ (TGF-β) and Interleukin-6 (IL-6)), which themselves act as OAFs, OIFs, or can directly promote further tumour cell proliferation and survival. For this reason, osteolytic bone disease is often described as a “vicious cycle” because the presence of tumour cells promotes osteolysis, and this further promotes tumour cell proliferation and survival. A number of adhesion molecules responsible for homing of MPCs to the bone matrix have been shown to directly support osteolysis (e.g., syndecan-1 [37] or vascular cell adhesion molecule-1 (VCAM1)) [38]. One such example is Notch, which is released when MPCs adhere to bone, and subsequent interactions with Jagged result in increased RANKL, driving OC differentiation [39,40]. MPCs also drive the release of OIFs (e.g., TGFβ, Dickopf-1 (DKK-1) and soluble Frizzled transmembrane receptors (sFRP2)), resulting in halted bone formation and therefore impaired bone repair. This is further perpetuated by sclerostin, another OIF, which is produced from OCYs and released at higher levels in the presence of MPCs. Ultimately, the catatonic state of MBD leads to net bone loss and reduced bone integrity and strength.

Current management strategies for MBD include: (1) pain relief, (2) interventional radiology (e.g., vertebroplasty), (3) orthopaedic interventions, and (4) bone targeted pharmacological treatments. Effective analgesia can be a challenge to optimise due to underlying renal impairment and susceptibility to bowel disturbances. Improving underlying bone damage, in partnership with pain management, is a preferable approach. Radiological procedures, such as vertebroplasty, can be helpful in the treatment of vertebral compression fractures. Orthopaedic interventions can either be prophylactic (for severe and/or unstable bone disease) or for fracture repair. The two approved pharmacological options for the treatment of MBD, bisphosphonates or denosumab (a monoclonal antibody inhibiting RANKL) [41], are both anti-resorptives and do little to drive the repair of existing bone lesions (Table 2). Improving our understanding of the impact of MPCs on OB physiology will allow us to explore novel targeted therapies to promote OB activity and the repair of cancer-induced bone damage. Here, we review the role of OBs in MBD and how this knowledge may drive future preclinical and clinical research, and, ultimately, disease management.

## 3. Osteoblast Dysfunction

As the only bone forming cell, OBs have a crucial role to play and account for approximately 5% of all bone cells in the BMME. In a physiological state, OBs would respond appropriately to OCY signalling, ensuring adequate bone formation in response to external stimuli. Having completed their physiological role, OBs either undergo apoptosis, differentiate into OCYs within the bone matrix, or differentiate into bone lining cells. OBs are mesenchymal stem cells (MSC) derived and committed to the OB lineage in the presence of transcription factors such as Runt-related transcription factor 2 (Runx2)/Cbfa1, β-catenin, and Osterix. MPCs both directly or indirectly oppose OB differentiation, function, and survival (Figure 2). In the presence of MPCs, OB precursors have depleted Runx2 expression [20] and overall reduced Runx2-positive OB numbers [23], leading to osteogenic suppression and reduced bone formation. Xu et al. have produced work suggesting that the presence of these Runx2 deficient immature OBs observed in MM can both attract and promote MPC progression within the BMME [42]. MPCs can also express Runx2, and when this has been observed clinically, it correlates to more aggressive disease phenotypes [23,43]. Trotter et al. demonstrated that Runx2 expressing MPCs develop some bone-cell phenotypes, which may allow them to reside in the BMME more effectively [43], thus perpetuating their impact on the bone. MM derived exosomes also exaggerate osteolysis, driving the release of IL-6 (to promote tumour growth) and inhibiting OB differentiation via reduced Runx2, Osterix, and Osteocalcin [44].

Adipocytes are also MSC derived, and work by Ruan et al. suggests that heparanase from MPCs may influence OB differentiation by high jacking OB lineage in favour of adipocytes, and that the mechanisms involve enhanced Peroxisome proliferator-activated receptor γ (PPAR-γ) expression, decreased Runx2, and increased DKK1 secretion [20]. This is supported by Liu et al., who show that MPC–MSC interactions (via integrin α4 and VCAM1) activate protein kinase Cβ1 to stabilise PPAR-γ2, which drives adipogenesis [45]. MPCs may also cause epigenetic changes in OBs, resulting in long-term dysregulation in bone remodelling [46]. One such pathway appears to be via growth factor independence 1 (Gfi1), a transcription repressor, which appears to be expressed by OB precursors in the presence of MPCs. Gfi1 binds to Runx2, causing a reduction in osteoblastic development. The inhibition of cell differentiation persists despite removal of MPCs from the BMME, explaining why persistent and new osteolytic disease is observed clinically for some patients [47]. Gfi1-induced OB inhibition has been shown in vitro to be reversible with anti-TNFα or anti-IL-7 treatments [17]. OBs also appear to be particularly susceptible to MM-induced apoptosis, mediated by Fas/Fas Ligand, TNF-α, and tumour necrosis factor-related apoptosis-inducing ligand (TRAIL), which is thought to be, in part, due to functional exhaustion in response to inflammatory cytokines [48]. Depleted OB numbers in MM (whether due to hindered osteogenesis, dysregulated function, or induced apoptosis) result in reduced total OPG, further exaggerating a bias towards bone loss via unopposed RANKL-driven osteoclastogenesis [49,50,51,52]. Here, we focus our review on some of the key factors inhibiting osteogenesis (Table 1).

## 4. Osteoblast Inhibiting Factors

### 4.1. Wnt/B-Catenin; DKK1, Frizzled Transmembrane Receptors and Sclerostin

Wnt/β-catenin signalling is the key regulator of OB differentiation and function, with multiple identified pathways (broadly categorised as canonical or non-canonical). Canonical Wnt signalling is β-catenin dependent, and when this pathway is activated, there is an expression of OB transcription factors. This canonical Wnt pathway is often inhibited in MM, leading to downregulated osteoblastogenesis (Figure 3) [53,54]. Non-canonical signalling occurs via the Frizzled (Fzd) receptor (in the absence of a co-receptor) and is β-catenin independent. Two key non-canonical cascades are the calcium-dependent pathway and the planar cell polarity pathway. Although non-canonical cascades are less characterised in MBD, Bolzoni et al. have recognised the importance of the non-canonical Wnt5a/ROR2 pathway, and that MPCs have the ability to inhibit ROR2 expression and downstream Wnt signalling. Additionally, the reactivation of this pathway supports osteogenic potential [55]. Many of the acknowledged OIFs identified in MM are thought to antagonise Wnt signalling (Table 1) [56,57]. Five key groups of secreted Wnt antagonist proteins are reported, including sFRPs, Dkk-1, Wnt inhibitory factor-1 (Wif-1), Wise, and Cerberus. Whereas sFRPs, Wif-1, and Cerberus are believed to bind to Wnt (thus preventing agonist-receptor interaction), Dkk-1 and Wise directly bind to low-density lipoprotein receptor related proteins (LRP5/6) to antagonise downstream signalling [56,58,59,60].

Dkk-1 antagonises Wnt/β-catenin via the binding of LRP-6 (co-receptors to Wnt) [61] and Kremen transmembrane proteins [62], inhibiting osteoblastogenesis by prevention of OB precursor differentiation into mature OBs. In 2007, MacDonald et al. demonstrated, using a hypomorphic DKK-1 mouse model, that lowering the expression of DKK-1 significantly altered the skeletal phenotype, resulting in thicker trabecular and cortical bone [63]. Dkk-1 is expressed from isolated human MM cells. Higher levels are seen in the BM and peripheral serum of patients with MM, with particularly elevated levels in MBD sufferers [14,15]. However, not all cases of advanced MBD are associated with the same upregulation of DKK-1 [15]. The expression of DKK-1 from MM cells has been shown to be particularly raised in some, but not all, subgroups of patients with specific MPC genetic profiles, indicating a link between DKK-1 and some genetic abnormalities [64]. In vitro exposure of OB cell lines to either recombinant Dkk-1 or co-cultures of MPCs (known to secrete Dkk-1) demonstrated altered OPG/RANKL secretion ratios favouring osteoclastogenesis, as well as the inhibition of Wnt [65]. Yaccoby et al. used a SCID-rab mouse model, implanting patient myeloma cells into bone chips treated with anti-DKK-1. Their findings showed an increase in OB numbers, a decrease in OC numbers, and an increase in BMD [66]. These preclinical studies suggest not only a benefit for patients with MBD, but also a reduction in MM cell growth [66,67]. An anti-Dkk-1 neutralising antibody, BHQ880, also yielded promising results preclinically and in early phase clinical trials. The 5T2MM murine myeloma model was treated with BHQ880, which prevented the development of osteolytic lesions [68]. Similar results were also seen in the INA-6 SCID-Hu murine myeloma model with increased OB numbers and trabecular bone numbers [69]. A phase Ib clinical study to assess BHQ880 in MBD resulted in increased BMD [70], but this study treated MBD concomitantly with zoledronic acid and anti-myeloma therapies. Assessment of BHQ880 as monotherapy in high-risk smouldering myeloma in a phase II single-arm study presented preliminary findings of bone anabolic activity radiologically [71]. Interestingly, some in vitro studies of anti-DKK-1 treatments suggested an antitumour effect, possibly mediated by a reduction in the MM growth factor IL-6 [66].

**Figure 3 jcm-10-03973-f003:**
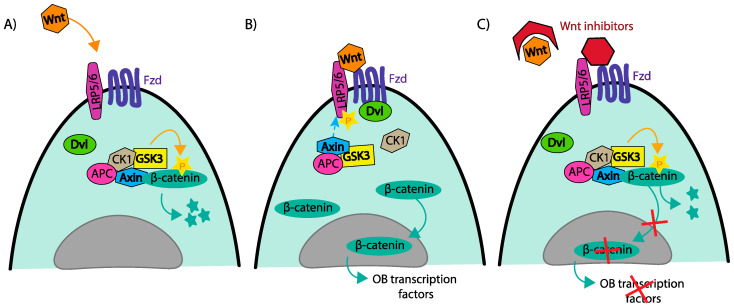
Canonical Wnt signalling in myeloma bone disease: (**A**) when canonical Wnt signalling is inactive, a destruction complex (including Axin, glycogen synthase kinase 3 (GSK3), adenomatous polyposis coli (APC), and casein kinase 1α (CK1α)) is responsible for continual phosphorylation and destruction of β-catenin. (**B**) When Wnt ligands bind to Wnt receptor Fzd and co-receptor LRP5/6, LRP5/6 becomes phosphorylated, which leads to recruitment of Dishevelled (Dvl), and, subsequently, Axin and GSK3 to the receptor complex [53]. As a result, there is a breakdown in the destruction complex. This leads to an increase in β-catenin translocation to the nucleus (due to downregulated phosphorylation), resulting in expression of OB transcription factors, and promotion of OB secretion of OPG [59,72]. (**C**) In MM, upregulated levels of Wnt inhibitors (red) antagonise Wnt either directly or indirectly (such as binding LRP5/6), leading to the downregulation of OB transcription factor expression.

sFRPs are glycoproteins, of which sFRP-2 and sFRP-3 are proposed to have a role in antagonising Wnt signalling in MBD (although sFRP-1 and sFRP-4 may also be involved) [58]. sFRPs act as decoy receptors to Wnt to prevent binding to Fzd-LRP5/6, with sFRP2 previously shown to be expressed by MPCs [33]. In vitro studies demonstrated inhibition of BMP-2-induced OB differentiation when cells were cultured in media from a variety of different myeloma cell lines known to express sFRP-2. When sFRP-2 was immunosuppressed, an increase in OB matrix mineralisation was also seen, suggesting sFRP-2 as a potential target for MBD treatment [33], viz., that it is possible to block and reverse its effects. Periostin, a cell-adhesion protein, also inhibits Wnt pathways, with elevated levels in MM associated with osteolysis [73].

Sclerostin is produced by OCYs and inhibits osteoblastic bone formation by antagonising canonical Wnt signalling [74], as well as preventing BMP-mediated OB mineralisation [75]. Increased levels of sclerostin are seen in BM samples from MM patients [31,76]. An extensive study of MPCs from 630 MM patients, and 54 MM cell lines, concluded that sclerostin is not expressed directly from the MM cells, but from OCYs [14]. Levels of sclerostin appear to be higher in more active MM disease states and fall post-chemotherapy during disease plateau phases [32], suggesting further clinical importance of the OCY in the context of MBD. In vivo studies assessing sclerostin as a therapeutic target for bone disease receive significant attention in the osteoporosis field. When assessing the treatment of anti-sclerostin antibodies in murine myeloma models, there was a subsequent increase in OB numbers, improved fracture resistance, and prevention of MBD development (including osteolytic lesions). In addition, there was an additive effect when treated concurrently with Zoledronic acid. [14]. Romosozumab is a monoclonal antibody against sclerostin, and is approved for the treatment of osteoporosis to improve BMD [77]. Phase III studies (NCT01796301 and NCT02186171) demonstrated that Romosozumab not only increased total hip and spine BMD [78,79], but also reduced new vertebral fracture rates by 48% when compared to the bisphosphonate Alendronic acid [80]. Unfortunately, Romosozumab was also associated with higher cardiovascular adverse events (2.5% vs. 1.9%) compared to Alendronic acid (NCT01631214) [80], but despite this, Romosozumab has now been approved for treating severe osteoporosis if patients do not have any cardiovascular risk factors. Other anti-sclerostin agents such as Blosozumab (LY2541546) are also in the clinical trial stages [81].

### 4.2. TGFβ, ACTIVIN A, BMPs, and HGF

Members of the TGFβ family control different processes of cell proliferation, differentiation, and apoptosis, as well as production of the extracellular matrix. TGFβ is a potent inhibitor of OB differentiation and function [82], and is activated and released from the bone matrix during osteoclastic resorption, as well as directly from MPCs [83]. Higher levels of TGFβ are observed in BM extracellular fluid samples from MM patients [84]. TGFβ knockout has been shown to have reduced bone mass and elasticity [85]. Paton-Hough et al. demonstrated in MM mouse models (JJN3 and U266) that inhibition of TGFβ with 1D11 (a monoclonal antibody) increased trabecular bone volume, BMD, vertebral strength, and repaired osteolytic lesions [86], with additional benefits when combined with anti-resorptive therapy. Nyman et al. have also assessed 1D11 in both immune-competent and immunocompromised murine models of MM and found improvements in bone volume, architecture, BMD, and vertebral bone strength [86,87]. A small molecule inhibitor to TGFβ receptor 1, SD208, has also shown in the non-MM C57BL/6 mouse model to have bone anabolic effects (increased trabecular bone volume, BMD, and OB activity), and these effects are thought to be driven by increases in cytokines such as Runx2 [88]. Green et al. demonstrated in NOD *scid* gamma (NSG) mice inoculated with human JJN3 myeloma cells that early treatment with SD208 prevented lytic lesion development. Subsequently, the group developed a low-tumour MBD mouse model (NSG mice inoculated with human U266-GFP-luc myeloma) and treated established lytic disease with chemotherapy with or without SD208. Treatment with SD208 improved bone structure, lesion repair, and fracture resistance when compared to chemotherapy alone in this established MBD model [89]. In vitro treatment of human myeloma BMSC samples with SD208 also enhanced OB differentiations [89]. Some TGFβ antagonists have also shown preclinical evidence of antitumour effects, in addition to their bone anabolic effects [36,90].

Activin A is a member of the TGFβ superfamily, and is also known to inhibit OB mineralisation, as well as drive osteoclastogenesis [67]. There is an association between increased Activin A levels and the presence of osteolytic lesions in MM patients [11]. Antagonising Activin A in vivo, in a myeloma murine model, prevented osteolytic lesion formation, upregulated OB activity, and increased BMD [12]. This was also confirmed in another study using Activin A inhibitor (RAP-011) [13]. Sotatercept (a recombinant activin type IIa receptor ligand trap (previously named ACE-011)) was developed as a bone anabolic. Early phase clinical studies in osteoporosis noted that Sotatercept increased haemoglobin levels and BMD compared to controls [71]. These studies directed trials assessing efficacy and safety in the treatment of anaemia, particularly in the context of renal failure and myelodysplasia [91,92]. A phase IIa trial, combining treatment of Sotatercept with melphalan, prednisolone, and thalidomide in MM patients, observed an increase in haemoglobin, BMD, and bone-specific alkaline phosphatase in all patients receiving Sotatercept (in the absence of bisphosphonates) [93]. Importantly, all patients receiving Sotatercept reported improved perceptions of MBD pain [93]. The increased haemoglobin and haematocrit observed in some studies may exclude the use of Sotatercept for patients with previous thrombotic events or polycythaemia.

Other members of the TGF-β family relevant in MBD are BMPs, with BMP-2 and BMP-7 having a role in the differentiation of OBs, and they are therefore of potential value as bone anabolic agents. BMP-2 has also been shown to have anti-proliferation effects on MPCs [94,95]. BMP signalling is dysregulated in MBD. In a 5TGM1 myeloma-bearing mouse model, blockade of BMP with a small molecule inhibitor to BMP type 1 receptor improved MBD outcomes (increased trabecular and cortical bone mass and decreased lytic lesions) [96]. Seher et al. treated MM cell lines in vitro with custom-designed BMP-2 variants and confirmed that there was an antagonistic effect on Activin A, as well as a BMP-2 agonistic effect, which could lead to overall net bone gain, as well as potential antitumour effects [95]. PIM kinases have been identified as potential targets for haematological malignancies, and also inhibit BMP-mediated osteoblastogenesis [18]. In vitro treatment of primary OC and OB cells from healthy donors with a pan-PIM kinase inhibitor can decrease OC formation and activity, as well as increase OB formation and function [97]. PIM inhibition in human MM murine models has resulted in reduced MBD [30,97]. A pan-PIM kinase inhibitor was assessed in early phase clinical trials in MM patients, but with a focus on anti-myeloma effects [98]. Hepatocyte growth factor (HGF) is expressed at higher levels in patients with MM compared to those with MGUS or no haematological pathology, with particularly elevated levels associated with lytic bone lesions [19]. HGF is released by BMSCs [99], OCs [100], and some MPCs [18,101], and has been implicated in MPC homing and survival in the BMME [102], as well as bone remodelling. Expression of the HGF receptor (cMet) on both OBs and OCs suggests a coupling regulation of this cytokine in bone remodelling, with HGF reducing osteoblastogenesis via BMP signalling pathways [18]. Treatment of human MSCs with HGF in vitro inhibited the BMP-2-induced expression of OB transcription factors Runx2 and Osterix, as well as downregulating Smad signalling (within the Wnt pathway), suggesting significant interference of osteoblastogenesis. This is supported by observations that increased HGF levels seen clinically in MM patients are associated with lower levels of serum markers of bone formation (bone-specific alkaline phosphatase and procollagen type 1 N-terminal propeptide) [18]. There is also some evidence to suggest that HGF has an indirect role as an OAF, as HGF appears to upregulate IL-11 production by OBs [26,103,104].

### 4.3. Interleukins

IL-3, IL-7, and IL-11 are expressed at increased levels within BM plasma in a high proportion of MM patients [21,26,105]. In vitro assessment has shown inhibition of early OB differentiation, with the treatment of both recombinant IL-3 and human BM plasma containing high levels of IL-3 [22]. IL-3 is thought to indirectly inhibit BMP-2-driven osteoblastogenesis via CD45+/CD11b+ haemopoietic cells [22]. An IL-3 receptor-targeting therapy, SL-401 (a recombinant diphtheria toxin and IL-3 drug [106]), was shown to promote OB formation in preclinical MBD studies [107]. IL-11 is frequently quoted as a known OIF in myeloma, but few research studies have focussed on its role. In non-myeloma in vivo studies, treatment with IL-11 has reduced bone formation [108], but also has a described role in promoting osteoclastogenesis [109]. In an oestrogen-deficient mouse, IL-7 has been shown to have anti-anabolic effects [24]. In the context of MBD, in vitro investigation has suggested that IL-7 drives the downregulation of Runx2-mediated osteoblastogenesis [23], and that immortalised BM MSCs, co-cultured with MM patient plasma, had a reduction in Runx2-positive OBs seen. Treatment with anti-IL-7 limited this inhibitory effect [110].

### 4.4. Other Factors

TNF-α can supress OB precursor cells as well as reduce Runx2, the result being depleted OB differentiation [34], thus downregulating OB formation in MM patients [35]. TNF-α also has pro-osteoclastic synergistic effects with very small levels of RANKL, and, therefore, may well also be an OAF in the context of MBD [111,112]. Another member of the TNF superfamily, LIGHT, also appears to antagonise OB differentiation, by reducing OB precursor formation and may also lead to sclerostin expression from monocytes [113]. MIP-1α, also known as chemokine cytokine ligand 3, is expressed at higher levels in MM patients with MBD. MIP-1α is a known OAF but is also responsible for inhibiting osteogenesis [28]. Depleted levels of Runx2, Osterix, and Osteocalcin observed with MIP-1α stimulation are partially reversed with an MIP-1α antibody [27,28].

## 5. Conclusions

The presence and growth of MPCs in the BMME frequently lead to the inhibition of osteogenesis, the promotion of OB apoptosis, and the dysregulation of OB function. As a result, the majority of MM patients develop MBD. Here, we have reviewed the impact of MM on OBs, with a focus on MM-induced factors known to inhibit and/or disrupt normal OB differentiation and function. OIFs have the potential to be targeted to promote osteoblastogenesis in MM patients, with the ultimate goal to repair bone, increase BMD, and improve quality of life. In particular, Romosozumab (which targets sclerostin) is now an approved therapy for the treatment of osteoporosis and could potentially provide a promising option for MBD treatment, pending further investigation in early phase clinical trials. Given the persistent suppression of OBs, despite successful cancer treatment, and the significant impact of disease, the field is long overdue therapeutic advances to support improved outcomes for patients with MBD [114].

## Figures and Tables

**Figure 1 jcm-10-03973-f001:**
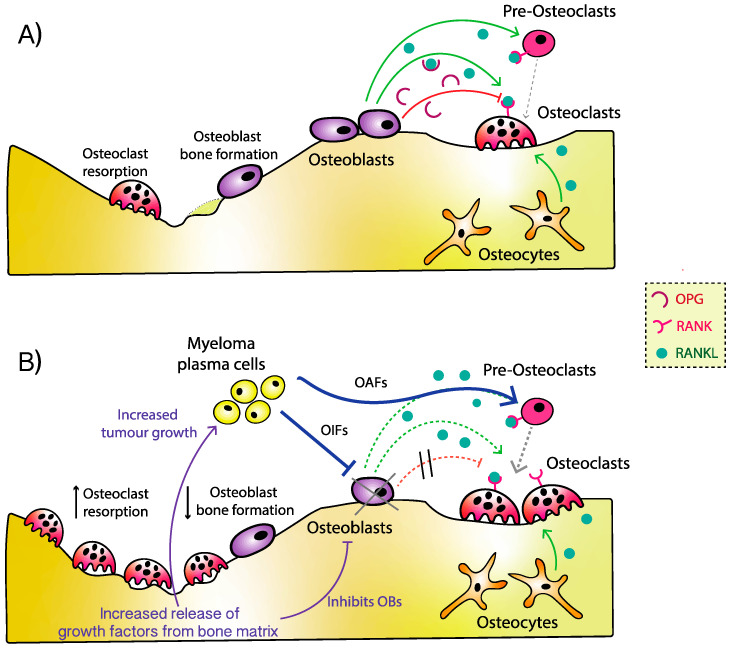
Myeloma bone disease: the vicious cycle: (**A**) in a disease-free state, bone remodelling is a balanced equilibrium of OC-driven bone resorption and OB-driven bone formation, controlled mainly by RANK/RANKL/OPG. OCYs and OBs release RANKL, which binds to RANK on pre-OC and OCs, promoting OC differentiation and activation. OBs also release OPG, which acts as a decoy receptor, and blocks RANKL to oppose osteoclastogenesis. (**B**) In the presence of MPCs, there are both directly and indirectly released OAFs and OIFs, resulting in a promotion of osteoclastogenesis and inhibition of osteogenesis. Bone resorption becomes unopposed, due to reduced OB numbers and depleted OPG. To add to this, increased levels of growth factors, such as TGFβ and IL-6, are released from the bone matrix, promoting tumour growth, as well as further opposing OB formation and function.

**Figure 2 jcm-10-03973-f002:**
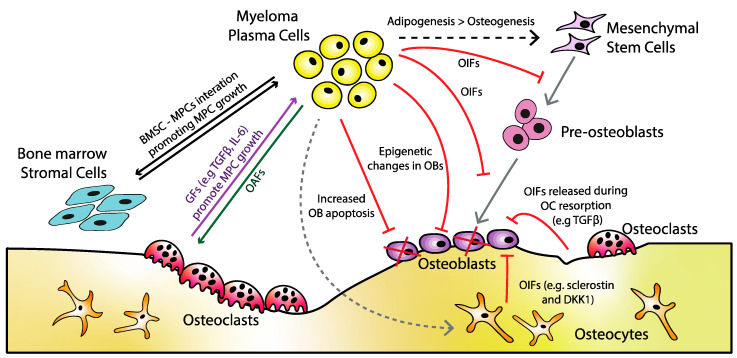
The inhibition of osteoblasts in myeloma bone disease. MPCs directly and indirectly promote OIFs that antagonise different stages of OB differentiation and maturation, as well as impacting on OB function. MPCs can also shift MSC differentiation towards a bias for adipogenesis (as opposed to osteogenesis), again deleting OB numbers. OBs exposed to MPCs are also more susceptible to apoptosis, as well as epigenetic changes that lead to long-term suppression of their function. The increased OC numbers and accelerated bone resorption seen in MBD also result in the release of growth factors (GFs) from the bone matrix, some of which further inhibit osteogenesis. MPCs have also been shown to increase the release of OIFs from OCYs, such as sclerostin. The GFs released during resorption, and MPCs adhesion to BMSCs, both enhance the homing and survival of MPCs, which further perpetuates OB inhibition and unopposed bone loss.

**Table 1 jcm-10-03973-t001:** Summary of osteoblast inhibiting factors.

OIFs	Expressed/Released by	Action	References
Activin A	OBs	Inhibits OB differentiation via SMAD2	[11,12,13]
Dkk-1	OB, BMSCs, and MPCs	Inhibits Wnt/β-catenin via LRP5/6 binding, increases osteoclastogenesis by reducing OPG	[14,15,16]
Gfi1	BMSC	Inhibits Runx2 expression	[17]
HGF	MPCs	Inhibits BMP signalling	[18,19]
HPSE	MPCs	Increases DKK1 (to inhibit Wnt signalling) and inhibits Runx2 expression	[20]
IL-3	BM T cells	Inhibits BMP-2 initiated OB differentiation	[21,22]
IL-7	BM T cells in MM	Decreases Runx2/Cbfa1 activity, inhibits OB differentiation/maturation	[23,24,25]
IL-11	Likely BMSCs	Dual role as OIF and OAF	[26]
MIP-1α (CCL3)	MPCs and macrophages	Inhibits Runx2 and downregulates Osterix	[27,28]
N-cadherin	MPCs	Over expressed in 50% MM patients, inhibits OB differentiation via inhibited Wnt signalling	[29]
PIM2	MPCs, MBSCs, and pre-OBs	Associated with reduced OB function, possibly via BMP2	[30]
Sclerostin	MPCs and OCYs	Inhibits Wnt/β-catenin via LRP5/6 binding, leading to inhibited osteoblastogenesis	[14,31,32]
sFRP-2	MPCs	Inhibits Wnt/β-catenin by altering Wnt/Frizzled binding (decoy receptor), inhibits BMP-2 induced OB differentiation	[33]
sFRP-3	MPCs	Inhibits OB differentiation via BMP-2	[33]
TNF-α	MPCs	Increases rates of mature OB apoptosis, possible due to interactions with Runx2	[34,35]
TGFβ	Bone matrix	Inhibits OB differentiation via Runx and DLX-5	[36]

**Table 2 jcm-10-03973-t002:** Summary of approved treatments for myeloma bone disease and investigational drugs in clinical trials.

Pharmaceutical Agents	Development Status	Mechanism	Action
Nitrogen-containing bisphosphonates (e.g., Zoledronate)	Approved in MBD	Inhibit farnesyl diphosphate synthase	Inhibit OCs
Non-nitrogen-containing bisphosphonates (e.g., Clodronate)	Approved in MBD	Inhibit ATP-dependent enzymes	Inhibit OCs
Denosumab	Approved in MBD	Anti-RANKL monoclonal antibody	Inhibit OCs
Romosozumab	Approved in OP, preclinical investigation in MBD	Anti-sclerostin monoclonal antibody	Promote OBs
Sotatercept	Phase IIa clinical trial in MBD	Recombinant activin type IIa receptor ligand trap	Promote OBs
BHQ880	Phase Ia and II clinical trials in MBD	Anti-Dkk-1 neutralising antibody	Promote OBs

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
