# Peer review of "Myeloma Bone Disease: The Osteoblast in the Spotlight"

_jcm, 2021, doi:10.3390/jcm10173973_

Round 1

Reviewer 1 Report

In the current review manuscript “jcm-1343726”, the authors highlight the role of bone modulating therapeutic agents that will enhance osteoblastic activity as a management approach in myeloma-induced bone disease.  The topic is generally of interest to the readers the suitability of JCM for this topic needs to be consulted. I have made a few suggestions that I believe will improve the overall quality of the manuscript.

  1. The manuscript does provide a good layout (short introduction, pathophysiology, mechanisms)
  2. MBD in the intro needs to be expanded.
  3. Figure 1 could be improved for clarity.
  4. The Solidarity of the assumptions made in the manuscript needs more scientific evidence at this stage.
  5. It would be valuable to include a table that summarizes all the current treatments approved or investigational drugs and their current stage with respect to their effects on osteoblast or osteoclast activities.

Author Response

We would like to thank the reviewer for their comments and suggestions. Please find below responses to the comments, point by point;

1) The manuscript does provide a good layout (short introduction, pathophysiology, mechanisms)

We are grateful that reviewer one likes the layout of the review, and agree that it works well. 

2) MBD in the intro needs to be expanded.

We thank the reviewer for pointing out that MBD had not previously been abbreviated, and this error has been rectified on line 14 of the introduction (with track changes left on for the editor)

3) Figure 1 could be improved for clarity.

Many thanks for this comment, there appears to have been a loss of quality to the figure, which we have tried to rectify by resaving the document without compression feature. I will discuss with the editor to ensure that the final version is clear, without the blurring that occurred in the originally submitted manuscript. We have also removed the labelling of OPG / RANK / RANKL from the schematic and instead added a key to make the figure clearer for the reader.

4) The Solidarity of the assumptions made in the manuscript needs more scientific evidence at this stage.

We appreciate the feedback from reviewer 1, and have changed the emphasis of the conclusions to ensure more balance. The final sentence of the abstract has also been reworded to reflect the reviewers comment. Wording has also been altered in sentence 7, paragraph 2 on page 3 and sentence 8 removed from the first paragraph of page 6.

5) It would be valuable to include a table that summarizes all the current treatments approved or investigational drugs and their current stage with respect to their effects on osteoblast or osteoclast activities.

We would like to thank the authors for this recommendation, and we have included an addition of a table of the current and early-phase investigational treatments for MBD. 

Reviewer 2 Report

The manuscript with the title “Myeloma Bone Disease; The Osteoblast in the Spotlight” is an interesting manuscript reviewing a large part of information of the relative field. The style of writing is comprehensible and the information is well organized. Only a few minor language corrections could be done.

Author Response

We would like to thank the reviewer for their comments and are very pleased that they find the review comprehensive and well organised. We thank reviewer two for suggesting that some minor language corrections may be needed, and we have endeavoured to address this.

  • An abbreviation had not previously been written in full within the introduction (MBD), this has now been corrected.
  • A duplicate full stop has been removed from page 3 paragraph 2.
  • A grammatical error has also been corrected in sentence 7, paragraph 2, on page 3.
  • An unnecessary abbreviation has been removed from the last paragraph of page 3.
  • The first paragraph of page 6, there have been some language corrections and a grammatical correction.

Corrections have been made to the resubmitted manuscript with track changes.